# Fluorescent Probes for Monitoring Toxic Elements from the Nuclear Industry: A Review

**DOI:** 10.3390/s25185835

**Published:** 2025-09-18

**Authors:** Clovis Poulin-Ponnelle, Denis Boudreau, Dominic Larivière

**Affiliations:** Radioecology Laboratory, Chemistry Department, Laval University, 1045 Avenue de la Médecine, Québec, QC G1V 0A6, Canada; clovis.poulinponnelle@cea.fr (C.P.-P.); denis.boudreau@chm.ulaval.ca (D.B.)

**Keywords:** sensors, fluorescence, monitoring, nuclear, uranium, cesium, strontium

## Abstract

With nuclear power playing an increasing role in efforts to reduce carbon emissions, the development of effective and sensitive monitoring tools for (radio)toxic elements in the environment has become essential. This review highlights recent advances in fluorescent probes developed for the detection of key elements associated with the nuclear industry, including uranium, cesium, strontium, technetium, zirconium, and beryllium. Various sensor platforms, ranging from organic ligands and DNAzymes to metal–organic frameworks and quantum dots, offer promising features, such as high sensitivity, selectivity, and suitability for environmental matrices. Several recent designs now achieve detection limits in the nanomolar to picomolar range, revealing new perspectives for environmental and biological applications.

## 1. Introduction

As the global community confronts the challenges of climate change and the urgent need to reduce greenhouse gas emissions, nuclear energy is increasingly recognized as a low-carbon, large-scale power source capable of complementing renewable energies [1,2,3]. Consequently, many countries have committed to expanding their nuclear energy capacity, resulting in the construction of new reactors and the extended operation of aging facilities. While nuclear power offers clear environmental and energy security advantages, it also raises critical concerns regarding the safe management of radioactive materials and their potential release into the environment [4,5]. Regulatory frameworks at national and international levels, such as IAEA or national nuclear safety agencies, demand the rigorous monitoring of radioactive contaminants across the nuclear fuel cycle, from mining to waste disposal [6]. As nuclear infrastructure ages and scales up, the development of efficient, rapid, and field-deployable detection methods becomes increasingly vital to ensure safety, regulatory compliance, and environmental protection.

Radionuclides originating from the nuclear industry are diverse and include fuel elements, fission products, and activation products [7,8]. Conventional pressurized water reactors use fuel containing uranium-238 (^238^U), enriched to approximately 3.5% with uranium-235 (^235^U). After the fuel is spent, it still contains about 95% uranium, with 1% of this being ^235^U. The spent fuel also contains about 1% plutonium, primarily ^239^Pu and ^240^Pu, and less than 0.1% minor actinides. Of these minor actinides, approximately half are neptunium-237, followed by americium-241, americium-243, and curium-244. Table 1 presents the main actinides found in the spent fuel, along with their characteristics.

The remaining 4% of the spent fuel consists of fission products, which are divided into two categories [9]. The first category includes medium-lived isotopes, with half-lives under 50 years, where cesium-137 and strontium-90 are the most prevalent. The second category comprises long-lived isotopes, with half-lives exceeding $10^{5}$ years and the major contributors being cesium-135, technetium-99, and zirconium-93. Table 2 presents the characteristics of the main fission products.

Activation products are materials that become radioactive after exposure to neutron radiation in a nuclear reactor. They originate from both the structural materials of the reactor (e.g., cobalt-60, iron-55, nickel-63, manganese-54, and beryllium-10) and the reactor coolant (e.g., nitrogen-16, tritium, carbon-14, and sodium-24) [10]. The nature and quantities of these activation products depend on factors such as the reactor design, the materials used in construction, and the reactor’s operating conditions, including neutron flux and exposure duration.

These elements present significant risks to human health and the environment due to their radiotoxicity and chemical behavior. Chemically, many of these substances tend to bioaccumulate in specific tissues, such as bones or organs. Their radioactivity, which deposits energy in tissues, can cause severe biological damage, such as cancer. This combination of radiological and chemical dangers necessitates safety measures to limit exposure and environmental contamination.

Monitoring environmental contamination using these elements is crucial for assessing and mitigating risks to ecosystems and human health. Various analytical techniques are employed for this purpose, including the development of sensors, compounds, or systems designed to interact with specific target ions and generate measurable responses. These sensors may be chemical, electrochemical, bacterial-cell, or protein-based, but they all rely on selective interactions with their target ions to produce a detectable signal [11]. This signal can manifest as a change in spectroscopic properties (such as absorption or fluorescence), electrochemical behavior (voltammetry, potentiometry), or even biological responses (such as gene expression). This review focuses on a specific class of optical probes: fluorescent sensors.

Fluorescent sensors offer significant advantages for environmental detection, including high sensitivity, rapid response times, and the capability for real-time monitoring [12,13]. These sensors can detect metal ions at low concentrations, often below the ppm level. Furthermore, sensitive and selective sensors reduce the need for complex sample preparation and expensive instrumentation, such as laboratory spectrometers. This makes them cost-effective tools for environmental monitoring, especially in portable devices, enabling direct on-site detection [14].

This review will focus on the fluorescent detection of key elements in the nuclear industry. Concerning transuranium elements, to the best of our knowledge, no fluorescent sensors have been developed for neptunium, plutonium, americium, or curium. The detection and quantification of these radioisotopes primarily rely on spectroscopic techniques such as alpha spectroscopy, liquid scintillation counting, UV-Vis-NIR absorption, Raman scattering, and mass spectrometry methods, including ICP-MS [15,16,17,18]. Thorium has limited use in the nuclear industry and has already been the subject of numerous reviews concerning fluorescent probes [19,20,21]. Fluorescent probes for uranium will be examined in detail, as it is the primary element used in nuclear applications.

As discussed previously, the main fission products are cesium with its two isotopes, followed by strontium, technetium, and zirconium. Therefore, their fluorescent probes will be detailed here. Activation products primarily include cobalt-60, iron-55, nickel-63, manganese-54, and beryllium-10. Except for beryllium, all of these are transition metals. Numerous reviews have already explored recent advancements in the development of fluorescent chemosensors specifically designed for detecting these transition elements [22,23,24,25]. To our knowledge, no review has specifically focused on fluorescent probes for beryllium, making its inclusion here necessary. This review focuses on the diversity of fluorescent probes, highlighting different sensor types, properties, matrices, and applications, and thus complements recent reviews on sensors for radioactive elements [26,27].

## 2. Fluorescent Sensors

Fluorescent sensors are categorized into three families based on their response to ion complexation: turn-on (fluorescence enhancement), turn-off (fluorescence quenching), and ratiometric (shift in emission wavelength) [28,29,30]. These responses are governed by different fluorescence mechanisms described in the following sections.

### 2.1. Photoinduced Electron Transfer

The photoinduced electron transfer (PeT) mechanism is commonly employed in the design of turn-on fluorescent probes. Such probes typically consist of a fluorophore linked via a spacer to a donor group able to complex the target ion with a certain level of selectivity (Figure 1). Upon absorption of a photoelectron, the electron occupying the highest occupied molecular orbital (HOMO) of the fluorophore transitions to the lowest unoccupied molecular orbital (LUMO) of the fluorophore. Subsequently, an electron transfer occurs from the HOMO of the donor group to the HOMO of the fluorophore, resulting in fluorescence quenching (Figure 1a). This process effectively blocks the emission transition of the excited fluorophore from the LUMO to the HOMO.

However, when the metal cation is complexed to the donor group, the energy level of the donor group’s HOMO becomes lower than that of the fluorophore’s HOMO (Figure 1b). As a result, the PeT in no longer possible, and the fluorescence from the probe is restored, enabled through the de-excitation of the fluorophore [31].

### 2.2. Photo-Induced Charge Transfer

Fluorophores that contain both an acceptor and a donor group facilitate the construction of ratiometric probes. Upon excitation, this kind of fluorophore undergoes an intramolecular charge transfer, known as a photo-induced charge transfer (PCT). The latter having an energy cost, the emitted photon will be of lower energy than the excitation photon, resulting in a red shift in the fluorescence spectrum compared to the absorbance spectrum. PCT depends of the dipole moment between the donor and acceptor groups in the fluorophore (Figure 2). Therefore, the interaction of a cation with one of these groups will change the dipole moment. Depending of the complexation site of the cation, the gap energy between LUMO and HOMO is changed, leading to a shift of the emission wavelength. If the donor site is involved, a blue shift (an increase in the energy gap) will result, whereas complexation via the acceptor site will lead to a red shift (a decrease in the energy gap). If the charge transferred is a proton, the effect is known as an excited state intramolecular proton transfer (ESIPT) [32].

### 2.3. Fluorescence Resonance Energy Transfer

Fluorescence emission can occur after resonance energy transfer (FRET) from a donor group to an acceptor group separated by a distance typically ranging from 10 to 100 Å. This non-radiative energy transfer occurs when the emission spectrum of the donor overlaps with the absorption spectrum of the acceptor. The excited donor transfers its energy via a long-range dipole–dipole interaction to the acceptor, which subsequently emits this energy as fluorescence.

This mechanism is widely employed in ratiometric fluorescent probes through two main designs. In the first, a metal ion can activate FRET by enabling or enhancing the fluorescence emission of either the donor or acceptor group (Figure 3a). This is achieved by increasing the spectral overlap between the emission spectrum of the donor and the excitation spectrum of the acceptor. In the second design, FRET is initially inhibited because the donor and acceptor groups are separated by a distance greater than 100 Å (Figure 3b). However, the binding of a metal ion can alter the molecular conformation, bringing the donor and acceptor closer together and thereby activating FRET.

### 2.4. Fluorescence Aggregation Mechanisms

Aggregation-induced emission (AIE) and aggregation-caused quenching (ACQ) are two contrasting photophysical phenomena that describe the behavior of fluorescent molecules in their aggregated states. In AIE, molecules are non-emissive in dilute solutions due to non-radiative decay pathways such as intramolecular rotations or vibrations but become fluorescent upon aggregation as these motions are restricted. Due to this property, AIE compounds have significant applications in bioimaging, sensors, and optoelectronics [33]. In the presence of specific ions, AIE sensors undergo aggregation, driven by ion coordination or interaction, which activates fluorescence.

Conversely, ACQ occurs when luminescent molecules lose their fluorescence upon aggregation due to strong intermolecular interactions like π-π stacking, which create non-radiative pathways such as excimer or exciplex formation. Although commonly observed in planar aromatic systems, ACQ is used to design turn-off fluorescent probes, which are often referred to as ’AIE turn-off’ probes in the literature.

### 2.5. Absorbance-Caused Enhancement

Absorbance-caused enhancement (ACE) is a fluorescence phenomenon through which the presence of an absorbing species near a fluorophore increases its light absorption, resulting in enhanced fluorescence emission. In ion sensors, ion complexation increases the absorbance of the complex, which in turn amplifies fluorescence and enables ion detection.

## 3. Fluorescent Probes for Nuclear Industry Elements

### 3.1. Uranium

Due to its natural abundance and significant role in the nuclear industry, uranium is the primary actinide studied for detection using sensors. A recent review by Wu et al. highlights that fluorescent sensors are the most popular choice, owing to their superior sensitivity and selectivity for uranium over other metals [34]. In nature, uranium is predominantly found in the +VI oxidation state, corresponding to the trans-dioxo form [UO_2_]^2+^, commonly known as uranyl. Uranyl ions accept ligands in their equatorial plane, typically with a coordination number ranging from 4 to 6 [35,36,37].

[UO_2_]^2+^ ions emit intrinsic fluorescence in the range of 450–600 nm under UV excitation. This property was widely exploited during the latter half of the 20th century for detecting uranium in geological samples [38]. The detection of trace levels of uranium requires laser excitation of the sample, typically using laser fluorimetry or, for enhanced sensitivity and speciation capability, time-resolved laser-induced fluorimetry [39]. The intrinsic fluorescence of uranium can be significantly enhanced by stabilizing it with an appropriate ligand. Dipicolinic acid (DPA), which forms a 1:2 complex with uranyl ions in aqueous medium ([UO_2_]^2+^:DPA), serves this purpose effectively [40,41]. DPA enhances the fluorescence emission of uranyl by an approximate factor of six compared to the emission in the absence of the ligand [42]. Trimesic acid and phosphoric acid have also been shown to enhance uranyl fluorescence in an aqueous solution [43,44]. Maji and Viswanathan further demonstrated a co-fluorescence effect with the addition of yttrium ions, enabling uranium detection at the ppm level [44].

However, the detection limits of these techniques, especially for field measurements, was proven to be limited. In response, the past two decades have seen a growing interest in the development of fluorescence sensors for [UO_2_]^2+^, significantly enhancing both sensitivity and selectivity.

#### 3.1.1. Organic Ligands

A wide variety of organic fluorescent sensors have been specifically developed for uranyl detection (Table 3). In most cases, a fluorescent moiety is linked to a coordination site for the [UO_2_]^2+^ cation. Uranyl complexation enables detection through one of three mechanisms: fluorescence enhancement (turn-on), fluorescence quenching (turn-off), or a shift in fluorescence emission.

A tetraphenylethylene (TPE) moiety (Figure 4a) linked to a uranyl-specific coordination unit has been widely utilized over the past decade for detecting uranyl ions (Table 3). TPE-based sensors are generally fluorescent, but their fluorescence is quenched upon uranyl complexation through the ACQ effect. Wen et al. reported the design of TPE-T, where the uranyl coordination unit is 2-(4,5-dihydrothiazol-2-yl)phenol [45]. In this system, uranyl ions are stabilized coordination with two TPE-T moieties, leading to ACQ. TPE-T exhibits 96% fluorescence quenching in the presence of uranyl ions, enabling clear differentiation from other metal ions. Additionally, TPE-T operates effectively across a broad pH range and demonstrates strong anti-interference capabilities, making it highly suitable for environmental applications, such as determining uranyl concentrations in river water.

**Table 3 sensors-25-05835-t003:** [UO_2_]^2+^ sensors. NMM: N-methyl-mesoporphyrin IX.

Type	Compound		Mechanism	$\lambda_{\mathrm{ex}}$/$\lambda_{\mathrm{em}}$ (nm)	LOD (nM)	Matrix	Refs.
TPE-	2-(4,5-dihydrothiazol-2-yl)phenol		ACQ	280/494	nd	River water (pH 2–10)	[45]
based	Salophen		ACQ	345/548	39	Drinking water (pH 3–9)	[46]
	Carbamoylphosphine oxide		ACQ	330/470	32	Tap water (pH 4)	[47]
	Amidoxime		ACQ	nd/444	4.7	River water (pH 4–7)	[48]
	Amidoxime		ACQ	340/485	7.9	Drinking water (pH 3–13)	[49]
	Carbodiimide		ACQ/PCT	346/512	0.07	Nuclear waste and seawater (pH 5–9)	[50]
	2-hydroxy-benzalaniline		AIE	350/500	11	Natural water (pH 4–9)	[51]
TPA-	2-(4,5-dihydrothiazol-2-yl)phenol		ACQ	350/510	50	Water (pH 4.5–7)/living cells	[52]
based	2-(Aminophenyl)iminomethylphenol		ACQ	380/550	39	Water (pH 2–10)/living cells	[53]
	TPA-benzoyl hydrazine		ACQ	384/510	0.2	Water (pH 2–10)/living cells	[54]
	4-aldehyde-4,4-bis(4-pyridyl)TPA		ACQ	430/525	0.01	Groundwater (pH 7.4)	[54]
Salicyl	4-pethoxycarboxyl SA		AIE	370/540	0.8	Nuclear wastewaters (pH 7–10)	[55]
aldehyde	3-hydroxy-flavone SA		AIE	370/457	2.1	H_2_O (pH 5–8.5)/living cells	[56]
azine (SA)	SA and 5-nitro SA		AIE	365/550	23	H_2_O/CH_3_CN mixture	[57]
Other	Calcein		Turn-off	492/520	60	H_2_O (pH 4)	[58]
organic	Quinoxalinol salen		Turn-off	450/540	nd	H_2_O/DMF mixture	[59]
ligands	Curcumin		Turn-off	424/507	nd	Tap water (pH 4)	[60]
	Esculin		Turn-off	390–455	6	H_2_O (pH 5–7)	[61]
	Naphthalimide		Turn-off	403/523	4100	Acetonitrile	[62]
	Furosemide		Turn-on	320/522	500	H_2_O (pH 5.5)	[63]
	Fluorophore	Quencher					
DNAzymes	Fluorescein	Black Hole	FRET turn-on	nd/520	0.05	Soil samples (pH 5.5)	[64]
	amidite	Quencher 1	FRET turn-on	487/520	0.02	Tap and river waters (pH 5.5)	[65]
	(FAM)	Dabcyl	FRET turn-on	nd/520	0.6	Living cells	[66]
		Graphene oxide	FRET turn-on	490/nd	0.03	Tap and river waters (pH 5.5)	[67]
		AuNPs	FRET turn-on	nd	0.01	Natural waters (pH 5.5)	[68]
		AuNPs	FRET turn-on	nd/520	0.0001	River water (pH 5.5)	[69]
		MoS_2_ nanosheets	FRET turn-off	495/525	0.002	River water (pH 5.5)	[70]
	Rhodamine	Guanine bases	FRET turn-on	399/609	0.09	Natural waters (pH 5.5)	[71]
	SYBR Green I	None	FRET turn-off	nd/525	0.2	Natural waters (pH 3.5)	[72]
Eu compound	Salophen-Eu-phosphate		PeT turn-on	no/413	8	Water (pH 4–9)	[73]
Eu compound	Eu-phosphate		PeT turn-on	400/597	10	Water (pH 7.2–8.8)/cells	[74]

Lin et al. introduced a second TPE-based fluorophore utilizing salophen as the uranyl coordination unit [46]. Salophen, N,N′-Bis(salicylidene)-1,2-phenylenediamine, is well known for its strong affinity for [UO_2_]^2+^, forming a stable complex with a 1:1 stoechiometry [75]. Similar to TPE-T, uranyl complexation with salophen results in fluorescence quenching, enabling a limit of detection as low as 39 nM. Lin et al. also developed a self-assembled monolayer incorporating TPE in conjunction with carbamoylphosphine oxide as the uranyl coordination unit [47]. These compounds are randomly distributed on a quartz glass substrate, enabling fluorescence quenching upon uranyl complexation. This configuration enhances detection capabilities, offering a robust platform for sensitive uranyl ion sensing.

Ding et al. and Zhan et al. developed uranyl sensors, TPE-A and TPE-ADX, respectively, based on TPE combined with amidoxime as the coordination unit [48,49]. Both sensors utilize fluorescence quenching upon uranyl complexation, facilitated via the interaction of two amidoxime moieties with the uranyl ion. TPE-ADX demonstrates a broader pH range (pH 3–13) compared to TPE-A (pH 4–7). Both sensors achieve low detection limits, with 4.7 nM for TPE-A and 7.9 nM for TPE-ADX. Feng et al. developed another TPE-based fluorescent sensor incorporating carbodiimide to create uranyl-specific coordination sites. This sensor achieves an ultra-low detection limit of 69 pM and demonstrates high selectivity by leveraging ACQ and PCT mechanisms. It operates effectively within a pH range of 5–9.

Lin et al. developed a distinct TPE-based sensor, in which TPE is linked to 2-hydroxy-benzalaniline, which operates through its hydrolysis induced via uranyl ions [51]. The hydrolysis product leads to enhanced and blue-shifted fluorescence upon interaction with [UO_2_]^2+^, contrasting with the fluorescence quenching observed in previous TPE-based sensors.

Triphenylamine (TPA) (Figure 4b) is widely used in the design of fluorescent sensors due to its strong electron-donating properties [76]. TPA-based sensors generally exhibit fluorescence, which is quenched upon interaction with uranyl ions, making them effective turn-off probes (Table 3). Zheng et al. designed a probe in which TPA is linked to a 2-(4,5-dihydrothiazol-2-yl)phenol coordination unit [52]. It exhibits high selectivity for uranyl ions, even in the presence of other metal ions, with interference from Cu^2+^ and Ni^2+^ being manageable through the use of masking agents like EDTA. Subsequently, Wu et al. developed a TPA-based probe, which incorporates a modified benzalaniline as the coordination unit for [UO_2_]^2+^ [53]. This sensor demonstrated a low detection limit of 39 nM, and it showed excellent anti-interference capabilities across a pH range of 2–10. Another TPA-based sensor, 4-aldehyde-4,4-bis(4-pyridyl)TPA, achieved an ultralow detection limit of 0.01 nM for uranyl in groundwater, relying on uranyl-triggered protein cleavage that ensures high selectivity by eliminating interference from other cations [54].

The salicylaldehyde azine (SA) moiety can also function as an AIE fluorophore (Table 3). Chen et al. demonstrated that 4-pethoxycarboxyl SA [55] and 3-hydroxy-flavone SA [56] function as turn-on AIE sensors. These compounds aggregate upon interaction with [UO_2_]^2+^, leading to fluorescence emission (Figure 5). Interestingly, SA and 5-nitro SA were shown to exhibit a contrasting behavior: uranyl ions quench the AIE effect in an organoaqueous solvent by disrupting aggregation, as reported by Pham et al. [57].

Nivens et al. demonstrated that the fluorescence of calcein, a dye molecule, is quenched upon complexation with uranyl ions [58]. Following the excitation of the complex at 425 nm, a photochemical reaction occurs, involving the decarboxylation of calcein and the dissociation of the complex. The photochemical oxidation reverses the quenching and enhances the fluorescence signal, making it a useful method for detecting uranyl ions selectively.

Various other organic ligands serve as turn-off probes, in which complexation with [UO_2_]^2+^ quenches fluorescence emission (Table 3). Among these are quinoxalinol salen ligands, for which the quenching is accompanied by a blue shift in emission, whereas Cu^2+^ complexation presents a red shift [59]. Curcumin forms a 1:2 complex with uranyl ions ([UO_2_]^2+^:curcumin), leading to a slight red shift in fluorescence and a distinct color change from bright yellow to orange [60]. The fluorescence quenching induced via uranyl complexation with esculin is further enhanced through the adsorption of esculin and uranyl ions onto SBA-15 resin [61]. Additionally, Kim and Tsukahara developed a sensor that combines diaza-18-crown-6 ether as the uranyl coordination site with a naphthalimide moiety as fluorophore [62].

On the other hand, the furosemide compound 4-chloro-2(furan-2-ylmethylamino)-5-sulfamoylbenzoic acid leads to an enhancement in the fluorescence emission by forming a 1:1 complex with uranyl ion [63].

#### 3.1.2. DNA

DNAzymes are DNA molecules with catalytic activity, often dependent on metal ion cofactors for their function. They are widely used in fluorescent sensors due to their ability to undergo conformational changes or catalyze specific reactions upon binding with target metal ions [77]. Liu et al. developed a DNAzyme with a selective complexation site for uranyl ions and a fluorophore positioned at the strand’s extremity [64]. The fluorescence emission of the fluorophore is suppressed by a quencher located opposite to it on the complementary strand, an effect attributed to FRET (Figure 6). A ribonucleotide adenosine (rA) site is incorporated near the uranyl complexation site; upon binding to uranyl ions, the DNAzyme catalyzes the cleavage of the rA site (optimal at pH 5.5 [70]), resulting in the release of the strand comprising the fluorophore. Once free from the quencher, fluorescence emission is recovered, enabling the detection of uranyl ions with a detection limit of 45 pM. Gold nanoparticles are integrated into this system to create biocompatible probes for detecting [UO_2_]^2+^ in living cells [78]. This model, based on the rA site and the release of a fluorophore, has been taken up by several other studies (Table 3).

Fluorescein amidite (FAM) remains one of the most commonly used fluorophores for designing turn-on probes, paired with various quenchers as detailed in Table 3 [64,65,66,67,68,69]. Notable examples include a dual probe for intracellular detection of Pb^2+^ and [UO_2_]^2+^ ions [66], the use of graphene oxide as a free quencher to avoid quencher labeling [67], and DNA amplification strategies to enhance sensitivity [68,69]. Turn-off probes can also be created, through which the quencher is not integrated into the DNAzyme. Instead, once released from the DNAzyme, FAM interacts with MoS_2_ nanosheets that quench its fluorescence [70].

Xiao et al. used tetramethyl-6-carboxyrhodamine as the fluorophore and four guanine bases as quenchers positioned at the strand’s extremity [79]. A PeT effect occurs between the fluorophore and the quenchers until the fluorophore is released, restoring fluorescence and functioning as a turn-on probe. N-methyl-mesoporphyrin IX is used as the fluorophore, while graphene oxide serves as a free quencher, to design a turn-on probe [71]. Zhu et al. developed a DNAzyme that, upon rA cleavage, undergoes a conformational change to form a G-quadruplex [72]. SYBR Green I, a fluorophore initially bound to the DNAzyme, loses its fluorescence when released, functioning as a turn-off probe.

The use of DNAzymes for [UO_2_]^2+^ detection via fluorescence demonstrates excellent limits of detection, with nearly all probes achieving sensitivity at the picomolar level (Table 3). Additionally, these probes exhibit good selectivity for uranyl and are often biocompatible, enabling their use for detection in living cells.

Other oligonucleotide-based sensors have been developed, utilizing fluorophores such as tryptophan [80] or dansyl groups [81] incorporated into a peptidic chain. Additionally, a salophen moiety has been immobilized onto silica gel particles as a solid-phase receptor for uranyl [82]. An oligonucleotide chain labeled with FAM subsequently interacts with the apical oxygen atom of the uranyl ion, forming a sandwich supramolecule. Following HCl elution, the fluorescence intensity is measured, allowing for uranyl detection with a LOD of 0.84 nM.

#### 3.1.3. Others

Jiang et al. developed a surface fluorescence sensor based on a Salophen-europium(III) complex for detecting uranium without external excitation [73]. The fluorescence mechanism is based on the cation–cation interaction (PeT process) between U(VI) and Eu(III) through a phosphate bridge, on the surface of a glass slide, leading to intense fluorescence with a LOD of 8 nM. This europium phosphate uranyl strategy has also been applied to the fluorescence imaging of [UO_2_]^2+^ in cells [74].

Several studies report the development of polymer-based sensors for the detection of uranyl ions via fluorescence quenching. The amidoxime group is frequently employed as a coordination site for uranyl within the polymer, enabling detection through the PeT effect [83,84]. In these systems, fluorescence is achieved by incorporating the amidoxime moiety into the polymer chain. Amidoxime groups can also be linked to 1,3,5-triethynylbenzene to create fluorescent microporous polymers, which allow the simultaneous adsorption and detection of uranium [85]. Additionally, other compounds such as porphyrin [86], trimetazidine [87], clopidogrel [88], and quinoline [89] have been incorporated into polymers to detect uranium via fluorescence quenching upon uranyl complexation.

Metal–organic frameworks (MOFs) exhibit high adsorption capacity for uranium. The incorporation of emissive central metal atoms enables luminescent properties, facilitating the development of MOF-based fluorescent sensors. A wide variety of MOF sensors has been developed, which includes different metal atoms such as Tb(III) [90,91,92,93,94,95], Co(II) [96], Zr(IV) [97], Ni [98], and Zn(II) [99,100]. These examples present limits of detection below 100 nM.

Covalent organic frameworks (COFs) have emerged as versatile fluorescent platforms for uranium detection and extraction, combining stability, tunable functionality, and rapid response [101,102,103,104,105,106]. Functionalized architectures such as *sp*^2^ carbon-conjugated amidoxime COFs [101], hydroxyl-functionalized 3D COFs [102], and β-keto-enamine COFs [106] enable selective sensing with low detection limits (typically 4–100 nM) in aqueous and environmental matrices. Further refinements include olefin-linked COFs bearing amidoxime, pyridine, and hydroxyl groups with improved recovery [103], exfoliated nanosheet COFs that mitigate aggregation quenching to achieve 10 nM sensitivity [104], and porphyrin-based COFs with sub-nanomolar limits of detection [105]. Collectively, these studies highlight the importance of rational functionalization and structural engineering in advancing COFs as regenerable probes for uranium monitoring in complex matrices.

Quantum dots (QDs) are extensively used as fluorescent probes, primarily in the form of carbon dots [107] and cadmium dots [108]. Carbon dots stand out due to their lower toxicity, which makes them particularly suitable for biological and environmental applications [109].

#### 3.1.4. Conclusions

The development of fluorescent sensors for uranium has made significant strides over recent decades. Researchers have designed various types of selective and sensitive probes, including organic ligands, DNAzymes, MOFs, and QDs. Advances in organic ligand design, especially in the last decade with TPE-based systems, have lowered detection limits into the nanomolar range, addressing environmental and industrial needs. Additionally, DNAzyme sensors containing the rA cleavage site have demonstrated detection limits in the picomolar range, with high sensitivity and biocompatibility, enabling applications in complex biological and environmental matrices. However, despite these promising results, key performance factors such as selectivity in real-world matrices, stability, and probe reusability are not often explored. These criteria must be addressed to ensure that fluorescent sensors can be reliably applied for the field monitoring of uranium contamination.

### 3.2. Cesium

Cesium-137 and Cesium-135 are medium-lived and long-lived fission products, respectively. ^137^Cs is more problematic due to its higher specific activity and greater decay energy. Among fission products, cesium poses significant problems after a release because it deposits in the soil and is readily absorbed by living organisms [110]. Its radioactive nature can cause severe health risks, including an increased risk of cancer, making its detection and monitoring crucial for environmental and public health safety [111].

Cesium ions present an oxidation state of +1, forming highly electropositive Cs^+^ ions. Due to its large ionic size and low charge density, Cs^+^ prefers forming complexes with bulky and non-polarizing ligands. The detection of cesium ions using fluorescent sensors has been extensively explored, leveraging various molecular designs and photophysical mechanisms. Kumar et al., in their review, pointed out two types of fluorescent sensors for Cs^+^: crown ether-based and calixarene-based crowns [112].

#### 3.2.1. Crown Ether-Based

Crown-ether-based fluorescent probes achieve ion selectivity through the size of their cyclic cavity and the nature of donor atoms [113]. Their strong, size-specific affinity for alkali and alkaline–earth metal ions allows integration into sensors to enhance both selectivity and binding efficiency (Figure 7a). When cesium ions interact with these crown ethers, the resulting complex induces a change in the photophysical properties of the attached fluorophore, often through the PeT mechanism.

Xia et al. demonstrated that a dicyano-substituted distyrylbenzene between 2 crown-6-ethers exhibits 1:1 coordination with Cs^+^, enhancing fluorescence with a selectivity for Cs^+^ greater than for K^+^ and Na^+^ [114] (Table 4). Otsuki et al. introduced two phthalimide units between the two crown-6-ethers, showing that 1:1 face-to-face coordination with Cs^+^ results in fluorescence quenching, whereas 1:2 coordination enhances fluorescence [115]. However, the selectivity of this sensor is higher for K^+^ and Na^+^ than for Cs^+^. Seo et al. combined an anthracene unit with a crown-5-ether, disabling the PeT effect for the Cs^+^ fluorescence detection, with a high selectivity for Cs^+^ than for other alkali metal ions [116]. Li et al. present a method for the adsorption and enrichment of Cs^+^ specifically using mesoporous silica containing dibenzo-24-crown-8-ether as the binding site for the alkali cation [117]. Here, the binding site does not rely on a fluorophore; instead, the addition of PbBr_2_ leads to the formation and growth of CsPbBr_3_ nanocrystals, which become fluorescent upon light excitation.

#### 3.2.2. Calixarene-Based Crowns

Calixarene-based receptors interact with a wide range of cations, with selectivity largely determined via cavity size, geometry, and donor atom type [135,136]. Alkali and alkaline earth metals typically bind through ion–dipole interactions with phenolic or ether oxygens, while transition and heavy metals form stronger coordination bonds with nitrogen, sulfur, or oxygen substituents. Lanthanides and actinides engage in multidentate interactions, through which their size and charge density play a crucial role in binding strength. Overall, subtle structural modifications of calixarenes enable fine-tuning of cation recognition, making them versatile platforms for selective sensing.

Concerning cesium ion, calix[4]arenes allow its coordination through electrostatic interactions. Particularly, 1,3-alternate calix[4]arenes with crown-6 rings (Figure 7b) are highly selective for cesium over sodium and moderately selective over potassium [137]. The crown-6 rings within these structures provide a robust and selective binding site for cesium, while the calix[4]arenes offer a versatile platform for attaching various fluorophores.

The fluorescence mechanism of these chemosensors depends on the fluorophore group attached to the calix[4]arene (Table 4). For example, cyanoanthracene [118,119,120,121,122], pyrene [123], and dansyl groups [125] rely on the PeT mechanism, while coumarin [124] operates through FRET. Dioxycoumarin [126,127,128,129] and BODIPY [130] fluorophores work through the PCT mechanism. To improve the solubility in water of these sensors, calix[4]arenes can be functionalized with sulfonate, carboxylate, or phosphate groups [127,129]. Additionally, modifying coumarin with cyano and benzothiazole groups can significantly increase the Stokes shift [128], enhancing the sensor’s performance and detection capabilities.

#### 3.2.3. Others

Other examples of chemosensors for Cs^+^ include a squaraine-based sensor that detects Cs^+^ ions through fluorescence quenching [132], a polyethylene glycol chain linking hydroxyphenyl and nitrophenyl moieties that utilizes the PCT effect [131], and a naphthalene-based macrocycle that becomes soluble and fluorescent upon binding to Cs^+^ [134]. Additionally, a sensor bearing imine and amide linkages, which provide binding sites for Cs^+^, can aggregate into fluorescent organic nanoparticles in aqueous media and emit fluorescence through the AIE effect [133].

#### 3.2.4. Conclusion

Two prominent types of sensors, crown-6-ether and calix[4]arenes, have been widely utilized for the selective detection of Cs^+^ ions. Both feature crown-6 rings that provide a highly selective binding site for cesium among alkali cations. Calix[4]arenes have become the primary scaffold for constructing these sensors. The diversity of fluorophores that can be attached to these binding sites influences key sensor characteristics, such as solubility and emission wavelength, allowing for adaptable detection strategies across a variety of environments. Indeed, the cesium detection described here is primarily performed in organic solvents due to fluorophore solubility constraints. However, a few studies have developed water-soluble fluorophores, such as calix[4]arenes bearing dioxycoumarin groups, allowing for the possibility of detecting cesium directly in natural water [127,129].

In addition, dual-function applications are possible, where fluorescent probes could aid both in the selective binding and in the preconcentration of radioactive cesium isotopes, thereby facilitating radiometric detection. This is particularly promising when such sensors are integrated into solid supports. By concentrating trace levels of cesium, these systems could enhance the ability to track radioactive isotopes of cesium in the environment [138].

### 3.3. Strontium

^90^Sr, the main radioactive isotope of strontium with a half-life of 28.8 years, is a product of nuclear fission in reactors and nuclear explosions. Due to its chemical similarity to calcium, it can replace calcium in bones, posing serious health risks [139]. Managing strontium-90 contamination involves limiting exposure and using chelating agents to reduce its presence in the body. Fluorescent probes for strontium detection are presented in Table 5.

#### 3.3.1. Fura-2

Fura-2, a widely used fluorophore in biology that contains a benzofuran group, was used to measure Ca^2+^ levels in tissue cells [140]. This fluorophore forms complexes with Ca^2+^, as well as other divalent cations such as Sr^2+^ and Ba^2+^, with nearly the same selectivity [141,142,143]. However, Fura-2, the first fluorophore studied for strontium detection, has not been further investigated for this purpose since the early 20th century.

**Table 5 sensors-25-05835-t005:** Strontium sensors.

Binding Site	Fluorophore	Mechanism	$\lambda_{\mathrm{ex}}$/$\lambda_{\mathrm{em}}$ (nm)	LOD (nM)	Matrix	Refs.
PEG chain	Pyrene	Excimer Formation	341/400	nd	Acetonitrile	[144]
Crown-ether	1,8-dioxyxanthone	PeT turn-on	308–314/424–437	nd	Methanol	[145]
	Merocyanine	PCT	399/612	nd	Acetonitrile	[146]
	BIC	ESIPT	369/414	40	H_2_O (5% DMSO)	[147]
			369/384	2000	H_2_O (NaLa micelle)	
Calixarene	Pyrene	Excimer Formation	325/397	100	Acetonitrile	[148]
	Quinoline	PeT turn-on	330/470	0.001	Waste water (pH 7) in acetonitrile	[149]
			284/575	1.4	Natural water (pH 7) in acetonitrile	[150]
	Vinylpyridinium	Turn-off	405/520	91	DMF/H_2_O (9:1)	[151]
G-quadruplex	Thiazole orange	Turn-on	480/535	10	H_2_O (pH 7)	[152]
	Thioflavin T	Turn-off	446/490	2.1	Natural water (pH 8.3)	[153]
FONs	Urea derivate	PeT turn-on	/380	190,000	Water (pH 2–7)	[154]
	Pyrene	Excimer Formation	345/480	9	Natural water (pH 2.7–11.2)	[155]

#### 3.3.2. PEG Chain and Crown Ether

Suzuki et al. demonstrated a probe consisting of two pyrene units connected via a PEG chain, where Sr^2+^ binding induces a conformational change [144]. The fluorescence, initially due to excimer formation of pyrene units, is shifted through monomer emission upon Sr^2+^ binding. A PEG chain with four or five ether units showed the strongest response.

As for cesium detection, crown ether-based fluorophores are also utilized for strontium detection. Prodi et al. utilized a 1,8-dioxyxanthone chromophore incorporated into crown ether structures of varying lengths, revealing that complexation with Sr^2+^, Ca^2+^, and Ba^2+^ significantly enhances fluorescence [145]. PEG chain length affects both binding affinity and fluorescence quantum yield. Plaza et al. employed a crown-ether-linked merocyanine fluorophore, observing a blue shift in emission upon Sr^2+^ binding due to the PCT effect, with notable selectivity over other ions such as Li^+^ and Ca^2+^ [146]. Akutsu-Suyama et al. reported another crown ether-based fluorophore, N-(2-hydroxy-3-(1*H*-benzimidazol-2-yl)-phenyl)-1-aza-18-crown-6-ether (BIC), which exhibits enhanced fluorescence in the presence of Sr^2+^ and shows selectivity over Na^+^, K^+^, Mg^2+^, Ca^2+^, and Ba^2+^ ions [147]. The detection of Sr^2+^ with this hydrophobic fluorophore was enabled in aqueous solutions by adding DMSO or using a sodium laurate micellar system.

#### 3.3.3. Calixarene

PyCalix is a fluorescent probe based on a calixarene scaffold featuring two pyrene moieties that exhibit intramolecular excimer emission [148]. Upon binding with Sr^2+^ cations, the separation of the pyrene units by Sr^2+^ leads to the suppression of intramolecular excimer emission and an increase in monomer fluorescence emissions accompanied by a blue shift. PyCalix shows high selectivity for Sr^2+^ and Ca^2+^ over other alkali and alkaline earth metals. However, its similar affinity for Ca^2+^ may interfere with the selective detection of Sr^2+^.

Sutariya et al. designed two calixarene-based compounds containing quinoline groups as the fluorescent units [149,150]. Both probes demonstrate high selectivity for Sr^2+^ over a wide range of other cations, with detection limits of 1 nM.

Fang et al. developed a sensor based on a calix[4]arene scaffold, which is modified to function as molecular tweezers through the incorporation of two vinylpyridinium iodide and two acetoxymethyl groups [151]. The sensor exhibits fluorescence quenching in the presence of Ca^2+^, Sr^2+^, and Ba^2+^, accompanied by red shifts in their UV-vis spectra, depending on the cation (570 nm for Sr^2+^). With its good cell permeability and water solubility, this sensor is a promising candidate for environmental monitoring and biological imaging applications.

#### 3.3.4. G-Quadruplex

G-quadruplex DNA structures, which are guanine-rich sequences stabilized by cations, are often used to detect strontium by luminescence due to their high selectivity for Sr^2+^ over other metal ions such as Ca^2+^, Na^+^, and Mg^2+^ [156,157,158]. Qu et al. developed a method for detecting strontium ions by fluorescence, utilizing G-quadruplex structures in combination with thiazole orange as the fluorophore [152]. The interaction of single-walled carbon nanotubes with the DNA strand leads to quenching of thiazole orange fluorescence. When Sr^2+^ ions induce the formation of G-quadruplexes, these structures dissociate from the nanotubes, resulting in fluorescence restoration. More recently, Feng et al. utilized the ability of thioflavin T to induce G-quadruplex formation in DNA to develop a switch-off probe for strontium [153]. Since Sr^2+^ has a higher affinity for G-quadruplexes than thioflavin T, the fluorescence intensity decreases with the addition of strontium, allowing for a ratiometric probe that achieves a detection limit of 2 nM.

#### 3.3.5. Fluorescent Organic Nanoparticles

Kaur et al. synthesized fluorescent organic nanoparticles (FONs) using the reprecipitation method with 1-(2-hydroxybenzyl)-3-naphthalen-1-yl-1-propyl-urea [154]. The binding of strontium ions to these FONs resulted in an increase in fluorescence due to the inhibition of the PeT process, achieving a detection limit of 2 ×$10^{2}$ µM. In a subsequent study, Kaur et al. introduced new FONs based on a pyrene-derived polymeric compound, which emitted fluorescence through an intramolecular excimer formation process [155]. The binding of strontium ions quenched this excimer formation, resulting in significantly higher sensitivity, with a detection limit of 9 nM for Sr^2+^. These FONs demonstrated high efficiency and selectivity for Sr^2+^ detection in aqueous media, showing promising practical applications in tap water, river water, and oral care products.

#### 3.3.6. Conclusion

Strontium detection using fluorescent probes has advanced significantly with the development of various types of sensors, incorporating different mechanisms and Sr^2+^ complexation sites, such as crown ethers, calixarenes, and G-quadruplexes. Initially, achieving selectivity over other divalent cations was a challenge, but this issue has been addressed over the past two decades, resulting in the development of selective probes. Contrary to cesium radioisotopes (^134^Cs and ^137^Cs), which can be readily detected via their γ-emissions, strontium-90, a β-emitter, is more challenging to detect in the field. The development of chemosensors could play a valuable role as screening tools for its rapid and selective detection.

### 3.4. Technetium

^99^Tc, an isotope of technetium, the lightest non-stable element, is a byproduct of nuclear reactors. This isotope raises substantial environmental concerns due to its long half-life and high mobility in environmental systems, especially in the pertechnetate form (TcO_4_^−^), which can readily migrate through soil and groundwater [159]. The perrhenate ion, ReO_4_^−^, is frequently employed as a non-radioactive surrogate of TcO_4_^−^ to study technetium detection through fluorescence due to its comparable physical and chemical properties. Fluorescent probes for technetium detection are presented in Table 6.

Arrigo et al. developed phosphiniminium cations linked to anthracene moieties (Figure 8a) that exhibit high selectivity for TcO_4_^−^, even in the presence of competing anions such as chloride, nitrate, and phosphate [160]. Although no significant spectral shift was detected, which limits its effectiveness as a fluorescent probe, the system demonstrated potential as a scintillation sensor for ^99^Tc detection due to its responsiveness to beta emissions from technetium’s radioactive decay.

Amendola et al. developed a fluorescent sensor for TcO_4_^−^ in water, using an azacryptand cage [161]. An anthracene incorporated into the azacryptand structure (Figure 8b) enhances the detection of TcO_4_^−^ through fluorescence quenching via the PeT effect in aqueous solutions at pH 2. This receptor is highly selective, showing no interference from environmentally prevalent anions such as chloride, nitrate, and sulfate.

The weak hydration properties of perrhenate and pertechnetate anions facilitate the formation of contact ion pairs with certain cationic fluorescent probes. Auramine O (Figure 8c), a cationic molecular rotor dye, forms fluorescent aggregates with ReO_4_^−^, resulting in both an emission enhancement and a red shift (AIE effect) [162]. Thioflavin-T and pseudoisocyanine (respectively, Figure 8d,e), other cationic fluorescent molecules, act as turn-on probes by aggregating with perrhenate anions [163,164]. Notably, pseudoisocyanine exhibits a particularly low limit of detection for perrhenate anions, reported at 0.2 µM, outperforming other sensors.

1-pyrenemethylamine (PMA, Figure 8f) is used as a turn-off probe with perrhenate anion [165]. The selectivity of the sensor is attributed to the strong hydrogen bonding and electrostatic interactions between a ReO_4_^−^ anion and the amino group of PMA, facilitating a photoinduced electron transfer process.

On the other hand, various macrostructures have been developed for the fluorescence detection of TcO_4_^−^ and ReO_4_^−^. Fluorescent MOFs are one such approach, with Rapti et al., who developed two Zr-based MOFs, MOR-1 and MOR-2, demonstrating effective sorption capacities for TcO_4_^−^ and ReO_4_^−^ [166]. Notably, MOR-2 functions as a selective luminescent sensor for ReO_4_^−^ in highly acidic conditions, where fluorescence quenching is based on a ligand-to-anion electron transfer. Xu et al. presented an example of a cationic photoactive porous aromatic framework material functionalized with an Ir(III) organometallic complex [167]. The detection mechanism for TcO_4_^−^ involves a strong, selective interaction between the Ir(III) complex and the anion, resulting in a fluorescence turn-on response. Li et al. described the coordination polymer TJNU-302, a cationic Ag(I) coordination polymer that incorporates a fluorescent ligand, 1,2,4,5-tetra(pyridin-4-yl)benzene [168]. TJNU-302 shows high selectivity and efficiency in capturing and sensing perrhenate ions, with fluorescence quenching attributed to hydrogen bonding between ReO_4_^−^ and the framework.

Choi and Lee developed a carbon quantum-dot-based sensor specifically designed for detecting perrhenate anions in aqueous solutions [169]. This sensor uses cationic carbon quantum dots functionalized with quaternary ammonium groups to enhance selectivity and sensitivity for ReO_4_^−^. The fluorescence mechanism relies on PeT process facilitated via electrostatic interactions between the cationic carbon quantum dots’ surface and the anions, resulting in fluorescence intensity reduction. Yi et al. introduced an ionic liquid-modified covalent organic framework incorporating nitrogen groups to bind ReO_4_^−^ anion [170]. The fluorescence quenching mechanism in this framework involves intramolecular charge transfer, upon ReO_4_^−^ binding.

Tc(CO)_3_^+^, a cationic form of ^99^Tc, is also present in significant amounts in nuclear waste. With the aim of monitoring Tc(CO)_3_^+^ in nuclear waste, Branch et al. developed a method to convert Tc(CO)_3_^+^ into a fluorescent complex using a bipyridyl ligand [171]. This complexation induces fluorescence, enabling the detection of Tc(CO)_3_^+^ with a limit of detection of 0.2 µM.

The fluorescence-based detection of technetium has progressed through the development of small-molecule probes and macrostructures designed to target the pertechnetate anion or its non-radioactive surrogate, the perrhenate anion in aqueous medium. These include organic ligands utilizing hydrogen bonding, MOFs, and carbon quantum dots achieving detection limits in the micromolar range.

### 3.5. Zirconium

Zirconium’s corrosion resistance, heat tolerance, and low neutron absorption make it ideal for cladding fuel rods in water-cooled reactors, with nearly 90% of global zirconium production dedicated to nuclear energy. Zirconium is commonly used in medical implants due to its biocompatibility and low chemical toxicity [172]. However, detecting zirconium could serve as a safety marker around nuclear reactors. The long-lived fission product ^93^Zr has low specific activity and radiation energy, resulting in minimal radiological risk. Zirconium is found in the environment in stable forms such as ZrSiO_4_ and ZrO_2_, which are largely insoluble in water, keeping zirconium fixed in soils and sediments [173]. Under certain conditions, zirconium can exist as Zr^4+^ ions or in various hydroxo and oxo complexes. Fluorescent probes for technetium detection are presented in Table 7.

The first published work appeared in 1951, when Alford et al. reported a fluorometric method for determining zirconium using 3-hydroxyflavone (Figure 9a) as a fluorescent probe [177]. This procedure involves forming a fluorescent complex between zirconium and 3-hydroxyflavone in a sulfuric acid solution, emitting blue fluorescence under ultraviolet light. The fluorescence intensity is directly proportional to zirconium concentration, enabling a quantitative analysis.

Sánchez et al. explored Ferron (8-hydroxy-7-iodo-quinoline-sulfonic acid, Figure 9b) in β-cyclodextrin under acidic conditions to detect zirconium at trace levels [178]. Encapsulation within cyclodextrin enhances fluorescence by shielding the excited species from quenching and nonradiative decay processes [183]. The hydrophobic cavity of cyclodextrin favors cation complexation, with Zr^4+^ complexation enhancing fluorescence, achieving a detection limit of 0.8 µM.

Mahapatra et al. introduced RhPT (Figure 9c), a rhodamine-based chemosensor, for detecting Zr^4+^ ions [174]. Upon zirconium binding, RhPT undergoes a structural shift from a spirolactam to a ring-opened form, which activates a delocalized xanthene tautomer within the rhodamine group. This transformation greatly enhances fluorescence and changes the naked-eye color from colorless to pink. The process is reversible, as demonstrated by the disappearance of the fluorescence upon the addition of excess EDTA. RhPT shows promise for live cell imaging, making it suitable for real-time monitoring of zirconium within complex biological systems. Sutariya et al. present a calix[4]arene-based fluorescent sensor (Figure 9d) specifically designed for the selective detection of Zr^4+^ and Fe^2+^ ions, thanks to its naphthalene units [175]. This sensor shows a significant fluorescence enhancement upon complexation with Zr^4+^, while complexation with Fe^2+^ inhibits fluorescence. In addition, other tested ions (Nd^3+^, La^3+^, Fe^3+^, Pr^3+^, Ce^3+^, Zn^2+^, Cd^2+^, Mn^2+^, Ca^2+^, Ba^2+^, Co^2+^, Hg^2+^, Ni^2+^, Pb^2+^, Sr^2+^, Cu^2+^, Li^+^, Ag^+^, Na^+^, K^+^, As^3+^) have no effect on the fluorescence. Selva-Kumar and Ashok-Kumar introduced a phenanthroline-based fluorescent probe (Figure 9e) designed for detecting Zr^4+^ in aqueous media [179]. This probe selectively binds zirconium ions with a 1:2 stoichiometry (probe:Zr). It exhibits an increase in absorbance at 412 nm upon interaction with Zr^4+^, accompanied by significant fluorescence enhancement via the PeT process.

Meng et al. developed a fluorescence turn-on probe for detecting zirconium ions using two complementary DNA strands [176]. Each strand is modified with pyrene on one end (3^′^ or 5^′^) and a phosphate group on the other. Upon binding with Zr^4+^, these oligonucleotides form a hairpin structure, bringing the pyrene molecules into close proximity and generating an excimer fluorescence signal. The introduction of a cyclodextrin further amplifies the fluorescence signal, enhancing the sensitivity of the probe, which achieves a detection limit of 2 ×$10^{2}$ nM. The probe exhibits high selectivity for Zr^4+^ over other metal ions, making it suitable for environmental and industrial applications.

A cobalt-based metal-organic framework was developed by Kirandeep et al. using a mixed-ligand approach that incorporates both N,N’-donor and polycarboxylate acid ligands to increase the interaction with the cations [180]. This MOF acts as a turn-on fluorescent sensor for Zr^4+^ via the absorbance-caused enhancement mechanism, demonstrating high selectivity and sensitivity with a detection limit of 67 nM. Moreover, MOF1 shows promise as a selective adsorbent for organic dyes, specifically Reactive Black 5 and Orange G. Chen et al. introduce a reusable Eu-based coordination polymer probe for Zr^4+^ detection, formulated as [Eu(L)_1.5_(phen)(H_2_O)] [181]. This probe is composed of 9,10-anthracenedicarboxylic acid and 1,10-phenanthroline, resulting in a complex with strong blue–violet fluorescence emission. The highly selective interaction with Zr^4+^ leads to fluorescence quenching, attributed to static weak interactions. Recently, Liao et al. developed fluorescent carbon quantum dots synthesized from o-phenylenediamine and L-cysteine [182]. These quantum dots exhibit high selectivity and sensitivity in detecting zirconium ions, with fluorescence quenching upon Zr^4+^ interaction. Additionally, they show a reversible pH response, with fluorescence quenching in acidic conditions (pH 1–5) and restoration in neutral or basic environments, making them suitable for pH-sensing applications.

Despite zirconium’s low toxicity, several fluorescent probes have been developed for its detection. Early systems such as 3-hydroxyflavone and Ferron operated in strongly acidic conditions and achieved only micromolar sensitivity, whereas modern probes demonstrate nanomolar detection in more practical matrices. The calixarene–naphthalene sensor offers the best sensitivity (1.4 nM) in river water, while phenanthroline (9 nM) and Co-MOF (67 nM) also perform well in aqueous environments. Carbon quantum dots combine good sensitivity (68 nM) with a broad pH tolerance and live-cell imaging capability. Importantly, the Eu-based coordination polymer achieves 200 nM detection limits while maintaining stability across pH 1–8, making it a strong candidate for reusable sensing platforms. Collectively, these examples show that zirconium probes can combine high sensitivity with adaptability to diverse matrices, supporting their potential in environmental monitoring, industrial safety, and biological applications.

### 3.6. Beryllium

Recognized for its exceptional physical properties, beryllium has been widely used in electronics and aerospace since the mid-20th century, owing to its high conductivity and thermal stability, and for its low density. Moreover, beryllium finds numerous applications in the nuclear industry. Thanks to its high neutron scattering cross section, it serves as a neutron reflector in nuclear reactors, enhancing the efficiency of reactions. Additionally, beryllium oxide acts as a neutron moderator, playing a crucial role in the control and regulation of nuclear reactions within the reactor core.

It is important to note that beryllium is highly toxic, posing significant health risks [184,185]. Classified as a carcinogenic element, it can lead to the development of chronic beryllium disease. Consequently, stringent measures must be implemented in industries dealing with beryllium to safeguard the health of employees. The development of fluorophores to monitor beryllium started in the 1950s, as scientists suspected an impact of beryllium on human health.

Morin (2′,3,4′,5,7-pentahydroxyflavone) emits a yellow-green fluorescence when it is complexed with beryllium in an alkaline solution [186], with the highest sensitivity at 0.2 M NaOH [187]. But Morin seems not to be selective since zinc, calcium, and lithium interfere with beryllium [188]. Additionally, three different complexes can form between beryllium and morin (M): BeM, Be_2_M, and Be_2_M_2_, depending on the pH, making it a complicated system for fluorescence detection [189].

The compound 10-hydroxybenzo[*h*]quinoline (HBQ) features a strong hydrogen bond between the hydroxy group and benzoquinolinic nitrogen. Owing to its π-electron system, it displays significant fluorescence [190,191,192,193]. The excitation of the fluorophore in its enol form induces the keto form through tautomerism with proton transfer (ESIPT process) from the hydroxy oxygen to the benzoquinolinic nitrogen. Subsequently, the excited-state keto form emits fluorescence before returning to the enol form through a reverse proton transfer. HBQ can also form a six-membered chelate ring with Be(II) by replacing the proton, causing a one-hundred-nanometer shift in the fluorescence emission maxima by inhibiting the ESIPT process. As HBQ is not soluble in water, its sulfonate derivative, HBQS (10-hydroxybenzo[*h*]quinoline-7-sulfonate), demonstrates good water solubility, making it a good probe for beryllium detection in solution (Figure 10).

Matsumiya et al. first developed an urban air analysis method using the fluorescence technique with HBQS in 2001 [194]. Following filter dissolution with nitric acid, beryllium is quantified in an alkaline solution (pH 12) through complexation with HBQS at a molar ratio of 1:1. The method achieved an LOD of 5 pg/cm^3^. Another method was developed to detect beryllium on surfaces using a filter-swiping technique [195]. Beryllium on the swipe is dissolved with ammonium bifluoride, before the addition of a detection solution, containing HBQS, EDTA as a masking agent, and L-lysine as buffer at pH 12.85. Building upon this approach, a portable fluorescence method has been established for the analysis of air and wipe samples with a limit of detection of 13.6 ng/swipe [196].

The close proximity between the hydroxy group and the benzoquinolinic nitrogen suggests that beryllium exhibits a high formation constant compared to larger metal cations. Additionally, alkaline conditions can lead to the hydrolysis of certain metal cations. EDTA is employed as a masking agent, given its weak equilibrium constant with Be(II) (10^−3.9^) compared to >10^8^ for other metal ions [197]. Consequently, no significant interference was observed from common metal cations, even with a molar excess of 10^5^ [194]. Fe and, to a lesser extent, TiO_2_ are the most significant interfering species, primarily due to the presence of hydrolyzed particles in suspension. However, this interference can be avoided by filtering the solution, which may exhibit visible color [195,196,198].

Matsumiya and Hoshino introduced a derivative of HBQ, (2-(2^′^-hydroxyphenyl)-10-hydroxybenzo[*h*]quinoline), used as a precolumn chelating reagent for beryllium in RP-HPLC [199]. The eluate is then detected via fluorometry with a detection limit of 39 fg/cm^−3^, and it does not present interferences from other metal ions.

HBQ was further functionalized onto a silicon nanopillar surface, onto which a 1 mL aliquot containing beryllium in a basic solution was applied [200]. Fluorescence emission, observed via microscopy, undergoes a shift when Be(II) is complexed with the Si-HBQ substrate (detection limit is 0.6 pg/cm^−3^). The presence of a thin layer of porous silicon oxide on the surface enhances the contact surface area, thereby amplifying fluorescence. The surface can then be rinsed with nitric acid to remove beryllium, enabling reuse of the substrate.

In addition to HBQ derivatives, a few fluorophores have been studied for beryllium detection. 4-methyl-6-acetyl-7-hydroxycoumarin forms a water-insoluble complex with beryllium, which can be extracted into benzene for measuring its fluorescence emission [201]. The concentration range for Be(II) is 0.5 to 10 ng/mL of benzene. The selectivity of this fluorophore for beryllium is influenced by the presence of citrate, EDTA, Ti, Zr, Hf, Cr, Zn, Sn, Sb, and Bi when their ratio compared to Be exceeds 2. Chromotropic acid and beryllium form a 1:1 complex (pH 4.8–6.0) that emits fluorescence, enabling beryllium detection within the range of 0.1 to 60 ng/cm^−3^ with no interferences reported among the sixty anions tested [202]. 2,6-diphenyl-4-benzo-9-crown-3-pyrane serves as a turn-on fluorescent probe for Be(II) in a MeOH/H^2^O (70:30, *v*/*v*) solution via the PCT process [203]. It exhibits a low LOD of 2 nM and shows no interference from tested metal ions.

Fluorescent probes for beryllium detection have significantly advanced, starting with Morin in the 1950s, despite its lack of selectivity. More sophisticated fluorophores like HBQ and its derivatives have since emerged, offering high sensitivity, low detection limits, and minimal interference from other metals. These advancements enable reliable beryllium monitoring in diverse environments while addressing its toxicity and environmental persistence.

## 4. Conclusions

Fluorescent probes have emerged as versatile tools for detecting toxic and radiotoxic elements relevant to the nuclear industry, including uranium, cesium, strontium, technetium, zirconium, and beryllium. Over the past decades, a wide range of sensing platforms, such as organic ligands, DNAzymes, MOFs, and quantum dots, have been developed. Recent breakthroughs, particularly with DNAzyme-based probes, COFs, and AIE systems, have pushed detection limits to the pico- and nanomolar range, enabling highly sensitive monitoring in both environmental and biological matrices.

These advances hold strong promise for real-world applications. Fluorescent probes can support environmental surveillance around nuclear facilities, assist in nuclear waste management, and track contamination from mining activities. In biological systems, several sensors have demonstrated compatibility with complex media, opening opportunities for studying radionuclide bioavailability, toxicity, and biodistribution. Importantly, fluorescence-based approaches provide a clear advantage for detecting alpha- and beta-emitting radionuclides, where radiometric techniques often require extensive sample preparation or lack sufficient sensitivity.

Despite these advances, several challenges remain before fluorescent probes can be reliably deployed in real-world settings. Improving selectivity in complex mixtures, enhancing sensor stability and reusability, and ensuring fluorophore resistance to both photodegradation and radiolysis are critical for long-term performance. Recent studies have highlighted strategies such as protective matrix embedding, the attachment of protective and anti-fading groups, and the optimization of the measurement medium as promising directions [204,205]. Future work should also emphasize the integration of these probes into portable devices, including microfluidic chips or point-of-care diagnostic platforms, while promoting cross-disciplinary research bridging chemistry, materials science, and radiochemistry. Addressing these issues will be essential to translate laboratory progress into robust, field-ready tools for nuclear safety, environmental monitoring, and biomedical applications.

## Figures and Tables

**Figure 1 sensors-25-05835-f001:**
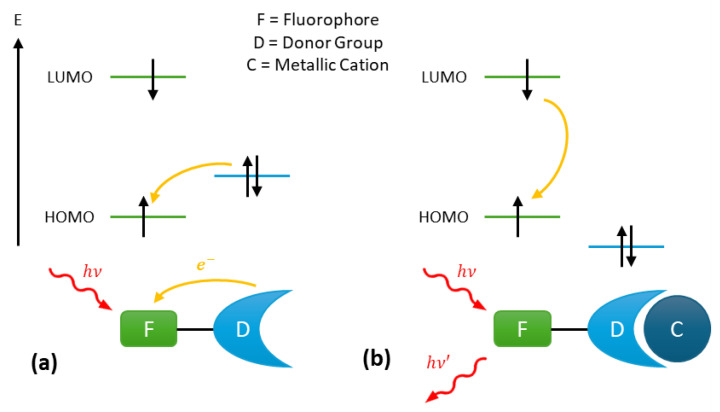
Fluorescence mechanism of photoinduced electron Transfer (PeT) probes: (**a**) fluorescence quenching by the donor group and (**b**) fluorescence enabling via cation complexation on the donor group.

**Figure 2 sensors-25-05835-f002:**
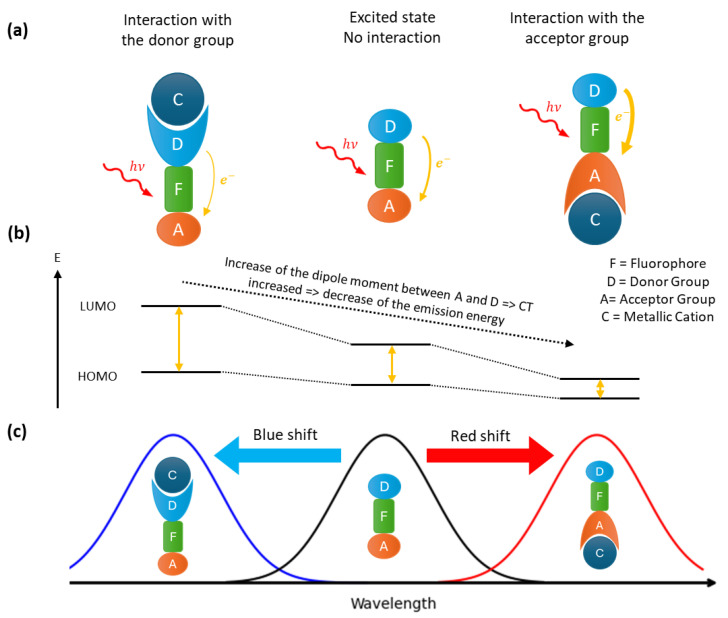
Fluorescence mechanism of PCT probes: (**a**) different interaction of the cation with the fluorophore and the corresponding spectral displacement (**b**) and fluorescence spectra (**c**).

**Figure 3 sensors-25-05835-f003:**
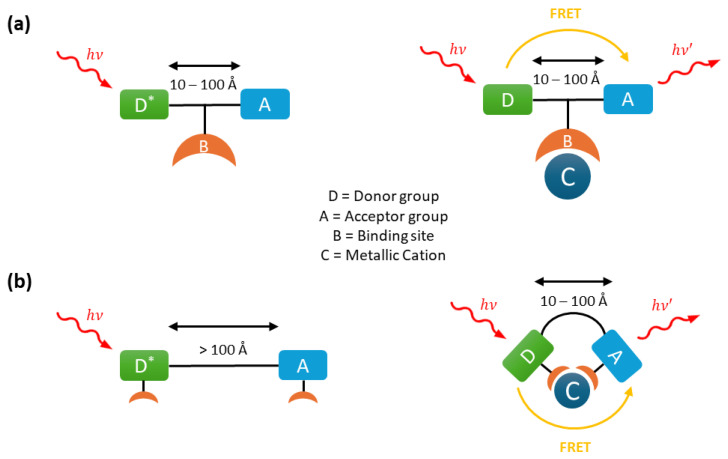
Fluorescence mechanism of Förster Resonance Energy Transfer (FRET) probes: (**a**) activation of FRET with cation binding and (**b**) with metal cation complexation bringing the two groups together.

**Figure 4 sensors-25-05835-f004:**
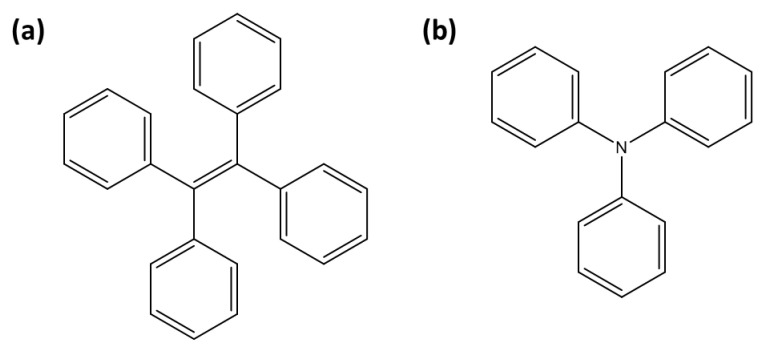
Representation of fluorescent moieties for [UO_2_]^2+^ detection: (**a**) tetraphenylethylene (TPE) and (**b**) triphenylamine (TPA).

**Figure 5 sensors-25-05835-f005:**
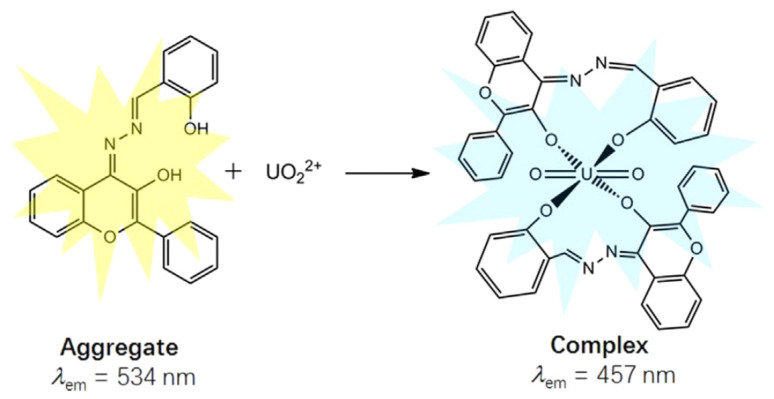
Schematic presentation for the ratiometric fluorescence change in 3-hydroxy-flavone salicylaldehyde azine upon binding with [UO_2_]^2+^ [56].

**Figure 6 sensors-25-05835-f006:**
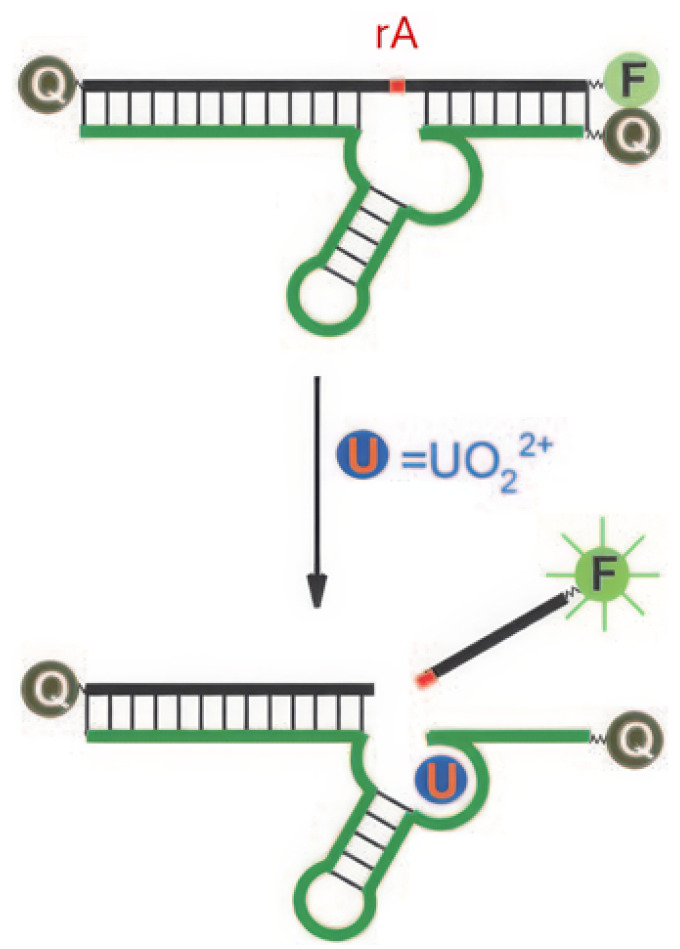
Principle of fluorescence detection of [UO_2_]^2+^ using a DNAzyme [64]. F represents the fluorophore, Q the quencher, and rA the ribonucleotide adenosine.

**Figure 7 sensors-25-05835-f007:**
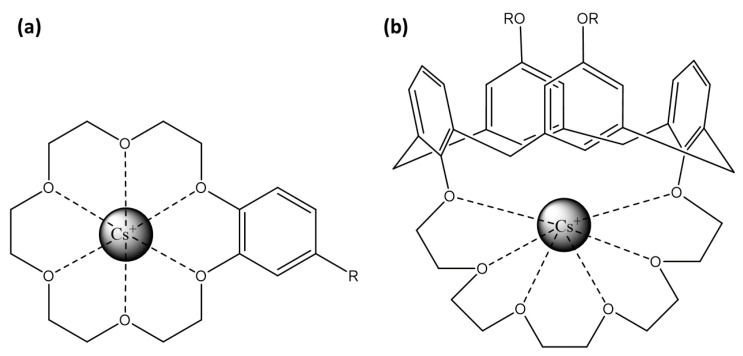
Representation of Cs^+^ binding site: (**a**) crown-6-ether and (**b**) calix[4]arene with crown-6 ring.

**Figure 8 sensors-25-05835-f008:**
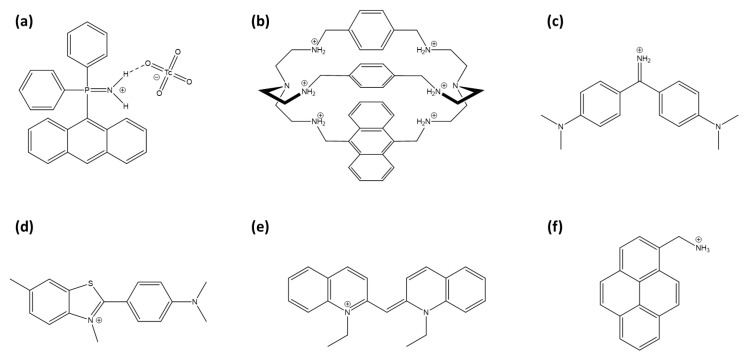
Representation of TcO_4_^−^ and ReO_4_^−^ sensors: (**a**) (9-anthracenyl)Ph^2^P=NH^2+^ cation, (**b**) modified azacryptand cage with anthracene, (**c**) auramine O, (**d**) thioflavin-T, (**e**) pseudocyanine and (**f**) 1-pyrenemethylamine.

**Figure 9 sensors-25-05835-f009:**
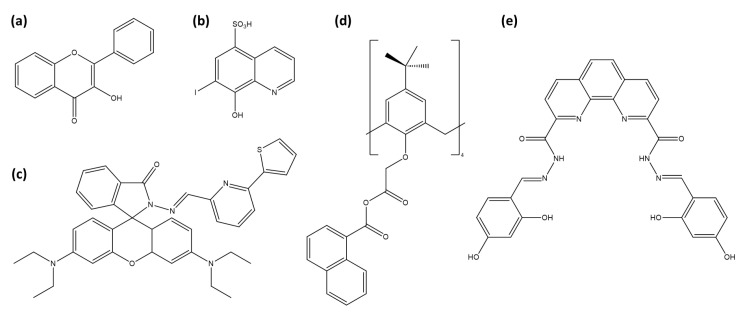
Representation of Zr^4+^ sensors: (**a**) 3-hydroxyflavone, (**b**) Ferron, (**c**) RhPT, (**d**) calix[4]arene with naphthalene units and (**e**) phenanthroline dicarbohydrazide based chemosensor.

**Figure 10 sensors-25-05835-f010:**
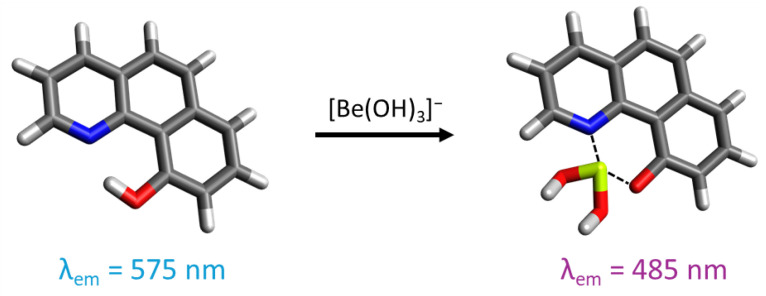
Representation of 10-hydroxybenzo[*h*]quinoline-7-sulfonate and its interaction with beryllium in an alkaline medium. For the illustration, the oxygen atoms are colored in red, the nitrogen atoms in blue, the sulfur atoms in yellow, the carbon atoms in gray, the beryllium atom in light green, and the hydrogen atoms in white.

**Table 1 sensors-25-05835-t001:** Actinides’ characteristics.

Isotope	Half Life Time (y)	Radioactive Decay	Energy (keV)
^238^U	4.5 × 10^9^	α	4269
^235^U	7.0 × 10^8^	αγ	4678
^239^Pu	2.4 × 10^4^	α	5 157
^240^Pu	6.6 × 10^3^	α	5168
^237^Np	2.1 × 10^6^	αγ	4957
^241^Am	433	αγ	5638
^243^Am	7.4 × 10^3^	αγ	5439
^244^Cm	18	α	5901

**Table 2 sensors-25-05835-t002:** Fission products’ characteristics. ^a^ Noble gas, no impact. ^b^ Decay product energy: 3.67 MeV.

Isotope	Half Life Time (y)	Radioactive Decay	Energy (keV)
medium-lived			
^137^Cs	30.08	βγ	514 + 662
^90^Sr	28.91	β	546
^151^Sm	90	β	76
^85^Kr ^a^	10.7	β	687
^155^Eu	4.7	βγ	147 + 87
long-lived			
^135^Cs	2.3 × 10^6^	β	269
^99^Tc	2.1 × 10^5^	β	298
^93^Zr	1.6 × 10^6^	β	91
^107^Pd	6.5 × 10^6^	β	34
^129^I	1.6 × 10^7^	βγ	149 + 40
^126^Sn ^b^	2.2 × 10^5^	βγ	378
^79^Se	3.3 × 10^5^	β	151

**Table 4 sensors-25-05835-t004:** Cesium sensors.

Binding Site	Fluorophore	Mechanism	$\lambda_{\mathrm{ex}}$/$\lambda_{\mathrm{em}}$ (nm)	LOD (µM)	Matrix	Refs.
Crown-6-ether	Distyrylbenzene	PeT turn-on	350/526	nd	Acetone	[114]
	Phthalimide	PeT turn-on/off	330/480–490	nd	Methanol	[115]
	Anthracene	PeT turn-on	400/500	3	H_2_O/CH_3_CN (1:1)	[116]
	CsPbBr_3_ NC	PeT turn-on	365/499–510	0.13	Natural water (pH 7)	[117]
Calix[4]arene	Cyanoanthracene	PeT turn-on	320–390/380–600	nd	Organic solvents	[118,119,120,121,122]
	Pyrene	PeT turn-on	360/380–420	nd	Ethanol	[123]
	Coumarin	FRET	245/420	nd	Acetonitrile	[124]
	Dansyl group	Ratiometric	330/541	nd	H_2_O/CH_3_CN (1:1)	[125]
	Dioxycoumarin	PCT	310/410	nd	Ethanol	[126]
			365/420	300	Water (pH 7)	[127]
			458/530	1400	H_2_O/CH_3_CN (2:8)	[128]
			365/420–438	800	Water (pH 7)	[129]
	BODIPY	PCT	500/530–580	nd	Acetonitrile	[130]
PEG chain	Phenols	PCT	365/500	nd	Solid samples	[131]
Squaraine		Turn-off	650/680	0.096	DMSO-H_2_O	[132]
Organic NP		PeT turn-on	309/412	0.070	H_2_O (pH 4–9)	[133]
			280/358	nd	H_2_O (pH 7.4)	[134]

**Table 6 sensors-25-05835-t006:** Technetium sensors. ^a^ 2-phenylpyridine, ^b^ 2,2 ^′^-bipyridine.

Binding Site	Fluorophore	Mechanism	$\lambda_{\mathrm{ex}}$/$\lambda_{\mathrm{em}}$ (nm)	LOD (µM)	Matrix	Ref.
Phosphinimine	Anthracene	nd	341/450	nd	Toluene	[160]
Azacryptand		PeT—turn off	377/425	nd	H_2_O (pH 2)	[161]
Auramine 0		AIE	490/560	270	H_2_O	[162]
Thioflavin-T		AIE—turn on	420/520	260	H_2_O (pH 7)	[163]
Pseudoisocyanine		AIE—turn on	550/640	0.2	H_2_O (pH 7)	[164]
1-pyrenemethylamine		PeT—turn off	339/375	14	H_2_O (pH 6–10)	[165]
MOF		PeT—turn off	360/460	1.5	H_2_O (pH 5)	[166]
Ir(ppy ^a^)2(bpy ^b^)^+^		PeT turn-on	365/645	5.6	Waste water (pH 4)	[167]
Ag(I) coordination polymer		PeT turn off	300/373	90	H_2_O (pH 3–13)	[168]
Carbon quantum dot		PeT turn off	254/540	87	Water	[169]
Ionic liquid		PCT	430/490	1.0	H_2_O (pH 2–12)	[170]
Bipyridyl		PeT turn on	415/573	0.2	Waste water (pH 4)	[171]

**Table 7 sensors-25-05835-t007:** Zirconium sensors.

Binding Site	Fluorophore	Mechanism	$\lambda_{\mathrm{ex}}$/$\lambda_{\mathrm{em}}$ (nm)	LOD (nM)	Matrix	Refs.
RhPT	Rhodamine	PeT turn-on	563/582	17,000	Living cells	[174]
Calixarene	Naphthalene	PeT turn-on	320/377	1.4	River water (pH 3–6)	[175]
Phosphate	Pyrene	Excimer Formation	344/474	200	Natural water (pH 7.4)	[176]
3-hydroxyflavone		PeT turn-on	UV/blue fluorescence	nd	0.2 N H_2_SO_4_	[177]
Ferron		PeT turn-on	365/490	800	H_2_O (pH 3.2)	[178]
Phenanthroline		PeT turn-on	340/518	9	DMSO/natural water (8:2) (pH 3–8)	[179]
Co-MOF		ACE	275/402	67	Tap water (pH 3.8)	[180]
Eu-based coordination polymer		Turn-off	370/415	200	H_2_O (pH 1–8)	[181]
Carbon quantum dots		PeT turn-off	390/560	68	H_2_O (pH 1–13)/living cells	[182]

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
