# Peer review of "Fluorescent Probes for Monitoring Toxic Elements from the Nuclear Industry: A Review"

_sensors, 2025, doi:10.3390/s25185835_

Round 1

Reviewer 1 Report

Comments and Suggestions for Authors

The manuscript provides a comprehensive overview of fluorescent probes for nuclear-related toxic elements with clear structure and sufficient references. However, some details need supplementation and clarification to enhance depth and practical guidance.
1. For zirconium sensors, supplement comparisons of detection limits and matrix adaptability among different probes to highlight their practical application potential.
2. Further explain the relationship between probe structures (e.g., calixarene derivatives) and their selectivity for target ions, with specific examples.
3. For specific cases, the response mechanisms are lacking in many of them and need to be supplemented.
4. Strengthen the discussion on challenges like sensor reusability and radiation resistance, and propose feasible solutions referenced in recent studies.
5. Standardize the format of detection limits in tables (e.g., unified units) to improve readability.

Author Response

The manuscript provides a comprehensive overview of fluorescent probes for nuclear-related toxic elements with clear structure and sufficient references. However, some details need supplementation and clarification to enhance depth and practical guidance.

  1. For zirconium sensors, supplement comparisons of detection limits and matrix adaptability among different probes to highlight their practical application potential.

We thank the reviewer for this valuable suggestion. We have revised the conclusion of the zirconium section to include a comparison of detection limits and matrix adaptability among the reported probes, highlighting their potential for practical applications.

“Despite zirconium’s low toxicity, several fluorescent probes have been developed for its detection. Early systems such as 3-hydroxyflavone and Ferron operated in strongly acidic conditions and achieved only micromolar sensitivity, whereas modern probes demon- strate nanomolar detection in more practical matrices. The calixarene–naphthalene sensor offers the best sensitivity (1.4 nM) in river water, while phenanthroline (9 nM) and Co-MOF (67 nM) also perform well in aqueous environments. Carbon quantum dots combine good sensitivity (68 nM) with a broad pH tolerance and live-cell imaging capability. Importantly, the Eu-based coordination polymer achieves 200 nM detection limits while maintaining stability across pH 1–8, making it a strong candidate for reusable sensing platforms. Col- lectively, these examples show that zirconium probes can combine high sensitivity with adaptability to diverse matrices, supporting their potential in environmental monitoring, industrial safety, and biological applications.”

  1. Further explain the relationship between probe structures (e.g., calixarene derivatives) and their selectivity for target ions, with specific examples.

An explanation of the interaction between both crown ether and calixarene and cations has been added:

“Crown-ether-based fluorescent probes achieve ion selectivity through the size of their cyclic cavity and the nature of donor atoms [113]. Their strong, size-specific affinity for alkali and alkaline-earth metal ions allows integration into sensors to enhance both selectivity and binding efficiency”

“Calixarene-based receptors interact with a wide range of cations, with selectivity largely determined by cavity size, geometry, and donor atom type [134,135]. Alkali and alkaline earth metals typically bind through ion–dipole interactions with phenolic or ether oxygens, while transition and heavy metals form stronger coordination bonds with nitrogen, sulfur, or oxygen substituents. Lanthanides and actinides engage in multidentate interactions, where their size and charge density play a crucial role in binding strength. Overall, subtle structural modifications of calixarenes enable fine-tuning of cation recognition, making them versatile platforms for selective sensing.”

  1. For specific cases, the response mechanisms are lacking in many of them and need to be supplemented.

We thank the reviewer for this constructive suggestion. In the revised manuscript, we have supplemented the description of the sensing mechanisms for several probes. Specifically, we added mechanism for DNAzyme-based sensors, Eu coordination compounds, and crown ether-based sensors, and zirconium and technetium sensors.

  1. Strengthen the discussion on challenges like sensor reusability and radiation resistance, and propose feasible solutions referenced in recent studies.

Reusability and radiation resistance is now discussed with references:

“Despite these advances, several challenges remain before fluorescent probes can be reliably deployed in real-world settings. Improving selectivity in complex mixtures, enhancing sensor stability and reusability, and ensuring fluorophore resistance to both photodegradation and radiolysis are critical for long-term performance. Recent studies have highlighted strategies such as protective matrix embedding, the attachment of protective and anti-fading groups, and the optimization of the measurement medium as promising directions.”

  • Goode, J.A.; Rushworth, J.V.H.; Millner, P.A. Biosensor Regeneration: A Review of Common Techniques and Outcomes. Langmuir 2014, 31, 6267–6276. https://doi.org/10.1021/la503533g.
  • Demchenko, A.P. Photobleaching of organic fluorophores: quantitative characterization, mechanisms, protection. Methods Appl. Fluoresc. 2020, 8, 022001. https://doi.org/10.1088/2050-6120/ab736

  1. Standardize the format of detection limits in tables (e.g., unified units) to improve readability.

The format of LOD was standardized, and all LOD values are expressed in the same molar unit.

Reviewer 2 Report

Comments and Suggestions for Authors

The abstract effectively introduces the topic of fluorescent probe development for detecting radionuclides and associated elements relevant to the nuclear industry. It highlights the importance of sensitivity, selectivity, and environmental applicability, while also pointing out advances in detection limits that enable broader applications. I would like to thank the authors for presenting an impactful review on the detection of radioactive metal ions using various fluorescent probes. However, the paper could be improved in the following ways:

  1. Authors selectively reviewed sensor platforms, ranging from organic ligands and DNAzymes to metal–organic frame works and quantum dots for the sensing of uranyl ions. Why don’t authors chosen the recent developed Covalent–Organic Framework based materials were discussed? 1021/acsmaterialslett.5c00777; 10.1080/03067319.2025.2490609; 10.1016/j.snb.2025.137564, etc.
  2. Several recent studies relevant to this topic have not been included in the present manuscript. For the review to be comprehensive and up to date, it is important that the authors incorporate and critically discuss these works. The suggested references are listed below. 1021/jacsau.5c00663; 10.1021/acs.inorgchem.5c01966; 10.1039/D5AY00379B; 10.3390/molecules30091920; 10.1016/j.saa.2025.126685; 10.1021/acs.inorgchem.4c05586; 10.1016/j.snb.2025.137643; 10.1007/s10967-024-09937-1; 10.1016/j.ica.2025.122655, etc.
  3. In the introduction part authors provided the data related to actinides characteristics, Fission products characteristics in the table 1 and 2 and explained without proper literature citation. I am recommending to authors to cite the relevant literatures.
  4. The authors briefly mention turn-on (fluorescence enhancement), turn-off (fluorescence quenching), and ratiometric (emission wavelength shift) sensing mechanisms. I recommend that this section be discussed in greater detail to enhance readers’ understanding. Furthermore, the inclusion of relevant citations : “10.1039/D3CP02714G; 10.1039/C5CS00496A; 10.1016/j.ccr.2025.216470; 10.1002/slct.202404525” will provide stronger support and context for these concepts.
  5. Table 3, which summarizes [UO₂]²⁺ sensors, is presented without a column outlining the probe stability, potential interferences of the reported studies, and the references also appear to be missing. I recommend that the authors include this information, as it would facilitate a clearer understanding of the methods and materials, highlight their respective strengths and limitations, and improve the overall utility of the table for both reviewers and readers.
  6. Tables 4 -7 require refinement to match the style and clarity demonstrated in Table 3, as previously recommended. In addition, the discussions related to Tables 4-7 explicitly integrated into the main text for better coherence and reader understanding.
  7. The conclusion section is overly lengthy and should be condensed to present the key findings in a clear and concise manner. A shorter, more focused conclusion will enhance readability and ensure that readers can easily grasp the main take home messages of the review.
  8. The authors should clearly articulate how the present review distinguishes itself from previously published reviews on similar topics. A comparative discussion highlighting the novel aspects, unique focus, or updated perspective of this work would help establish its added value to the field. For example, 10.1039/D5AN00683J; 10.1007/s00604-025-07308-5.

Author Response

The abstract effectively introduces the topic of fluorescent probe development for detecting radionuclides and associated elements relevant to the nuclear industry. It highlights the importance of sensitivity, selectivity, and environmental applicability, while also pointing out advances in detection limits that enable broader applications. I would like to thank the authors for presenting an impactful review on the detection of radioactive metal ions using various fluorescent probes. However, the paper could be improved in the following ways:

  1. Authors selectively reviewed sensor platforms, ranging from organic ligands and DNAzymes to metal–organic frame works and quantum dots for the sensing of uranyl ions. Why don’t authors chosen the recent developed Covalent–Organic Framework based materials were discussed? 1021/acsmaterialslett.5c00777; 10.1080/03067319.2025.2490609; 10.1016/j.snb.2025.137564, etc.

We have added COF sensors to our review:

“Covalent organic frameworks (COFs) have emerged as versatile fluorescent platforms for uranium detection and extraction, combining stability, tunable functionality, and rapid response [99–104]. Functionalized architectures such as sp2 carbon-conjugated amidoxime COFs [99], hydroxyl-functionalized 3D COFs [100], and β-keto-enamine COFs [104] enable selective sensing with low detection limits (typically 4–100 nM) in aqueous and environmental matrices. Further refinements include olefin-linked COFs bearing amidoxime, pyridine, and hydroxyl groups with improved recovery [101], exfoliated nanosheet COFs that mitigate aggregation quenching to achieve 10 nM sensitivity [102], and porphyrin-based COFs with sub-nanomolar limits of detection [103]. Collectively, these studies highlight the importance of rational functionalization and structural engineering in advancing COFs as regenerable probes for uranium monitoring in complex matrices.”

  • Cui, W.R.; Zhang, C.R.; Jiang, W.; Li, F.F.; Liang, R.P.; Liu, J.; Qiu, J.D. Regenerable and stable sp2 carbon-conjugated covalent organic frameworks for selective detection and extraction of uranium. Nat. Commun. 2020, 11. https://doi.org/10.1038/s41467-0 20-14289-x.
  • Cui, W.R.; Chen, Y.R.; Xu, W.; Liu, K.; Qiu, W.B.; Li, Y.; Qiu, J.D. A three-dimensional luminescent covalent organic framework for rapid, selective, and reversible uranium detection and extraction. Sep. Purif. Technol. 2023, 306, 122726. https://doi.org/10.1016/ j.seppur.2022.122726.
  • Zhen, D.; Liu, C.; Deng, Q.; Li, L.; Grimes, C.A.; Yang, S.; Cai, Q.; Liu, Y. Novel Olefin-Linked Covalent Organic Framework with Multifunctional Group Modification for the Fluorescence/Smartphone Detection of Uranyl Ion. ACS Appl. Mater. 2024, 16, 27804–27812. https://doi.org/10.1021/acsami.4c05522.
  • Xu, D.; Lu, S.; Hao, X.; Yang, S.; Lu, J.; Huang, Y.; Chen, W.; Hao, H.; Huang, S.; Chen, L.; et al. Highly Sensitive Detection of ppb-Level Uranyl via Exfoliation-Activated Luminescent Covalent–Organic Framework Nanosheets. ACS Materials Letters 2025, 7, 2716–2724. https://doi.org/10.1021/acsmaterialslett.5c00777.
  • Guo, Z.; Huang, X.; Mai, Z.; Yang, L.; Zheng, S.; Liao, J.; Gao, F.; Zhang, Y.; Jiao, Z. Development of COF-MF-based fluorescent sensor for on-site, and portable detection of uranium in ore samples. Int. J. Environ. An. Ch. 2025, pp. 1–12. https: //doi.org/10.1080/03067319.2025.2490609.
  • Liu, Y.; Liu, C.; Deng, Q.; Yu, Y.; Tang, X.; Li, L.; Grimes, C.A.; Yang, S.; Cai, Q.; Zhen, D. A β-keto-enamine covalent organic framework fluorescent switch for selective and sensitive UO2+ 2 detection. Sens. Actuators, B 2025, 433, 137564. https: //doi.org/10.1016/j.snb.2025.137564.

  1. Several recent studies relevant to this topic have not been included in the present manuscript. For the review to be comprehensive and up to date, it is important that the authors incorporate and critically discuss these works. The suggested references are listed below. 1021/jacsau.5c00663; 10.1021/acs.inorgchem.5c01966; 10.1039/D5AY00379B; 10.3390/molecules30091920; 10.1016/j.saa.2025.126685; 10.1021/acs.inorgchem.4c05586; 10.1016/j.snb.2025.137643; 10.1007/s10967-024-09937-1; 10.1016/j.ica.2025.122655, etc.

Thanks for proposing these references. Recent studies conducted since the end of 2024 have been added. The references you cited have been added and discussed in the different sections of the review (MOF, TPA-based, and Eu composite sensors).

“Another TPA-based sensor, 4-aldehyde-4,4-bis(4-pyridyl)TPA, achieved an ultralow detection limit of 0.01 nM for uranyl in groundwater, relying on uranyl-triggered protein cleavage that ensures high selectivity by eliminating interference from other cations.”

  • Chen, Y.; Yan, Z.; Zhang, D.; Zhao, X.; Guo, Z.; Huang, X.; Qu, J.; Zhang, J.; Wang, Y.; Jiao, Z. Highly selective and portable fluorescent detection of uranyl in groundwater via AIEgen-labeled protein cleavage. Spectrochim. Acta, Part A 2026, 344, 126685. https://doi.org/10.1016/j.saa.2025.126685.

“Jiang et al. developed a surface fluorescence sensor based on a Salophen-europium(III) complex for detecting uranium without external excitation [72]. The fluorescence mechanism is based on the cation-cation interaction between U(VI) and Eu(III) through a phosphate bridge, on the surface of a glass slide, leading to intense fluorescence with a LOD of 8 nM. This europium phosphate uranyl strategy has also been applied to fluorescence imaging of [UO2] 2+ in cells [73].”

  • Jiang, M.; Xiao, X.; He, B.; Liu, Y.; Hu, N.; Su, C.; Li, Z.; Liao, L. A europium (III) complex-based surface fluorescence sensor for the determination of uranium (VI). J. Radioanal. Nucl. Ch. 2019, 321, 161–167. https://doi.org/10.1007/s10967-019-06566-x.
  • Zhou, X.; Wang, Y.; Xiong, L.; Song, J.; Zhou, H.; Li, L.; Zhen, D. Development of rare earth europium composites for highly sensitive fluorescence enhancement for detection of uranyl ions in water and cells. J. Radioanal. Nucl. Ch. 2024, 334, 1931–1939. https://doi.org/10.1007/s10967-024-09937-1

MOF sensors publication addition:

  • Xie, J.; Liang, J.; Lei, J.; Xiao, Y.; Luo, F.; Hu, B. Highly Sensitive and Selective Detection of Uranyl Ions Based on a Tb3+- Functionalized MOF via Competitive Host–Guest Coordination. Inorg. Chem. 2025, 64, 3616–3625. https://doi.org/10.1021/acs.inorgchem.4c05586
  • Li, S.Q.; Peng, H.B.; Cao, X.H.; Dong, Z.M.; Wang, Y.Q.; Zhang, Z.B.; Liu, Y.H. Defect-engineered UiO-66-NH2 (AO)-3: Achieving enhanced fluorescence sensing of uranyl ions. Inorg. Chim. Acta 2025, 581, 122655. https://doi.org/10.1016/j.ica.2025.122655
  • Sun, Y.; Yu, L.; Wu, K.; Yin, M.; Lu, Y.; Yuan, Z.; Jiang, W.; Wang, J.; Wang, X.; Wang, S. Non-rare earth doped metal-organic framework for fluorescent detection of uranyl in real seawater. Sens. Actuators, B 2025, 436, 137643. https://doi.org/10.1016/j.snb.2025.137643.

  1. In the introduction part authors provided the data related to actinides characteristics, Fission products characteristics in the table 1 and 2 and explained without proper literature citation. I am recommending to authors to cite the relevant literatures.

Some literature has been added to the introduction:

  • Bruno, J.; Ewing, R.C. Spent Nuclear Fuel. Elements 2006, 2, 343–349.
  • Tomar, B.S. Nuclear Fuel Cycle, 1st ed. ed.; Springer: Singapore, 2023.
  • Nawada, H.P. Status of minor actinide fuel development; Number 1411 in Publication, Internat. Atomic Energy Agency: Vienna, 2009.
  • Nagasaki, S. Radioactive Waste Engineering and Management; Number v.6 in An Advanced Course in Nuclear Engineering Ser., Springer Japan: Tokyo, 2015

  1. The authors briefly mention turn-on (fluorescence enhancement), turn-off (fluorescence quenching), and ratiometric (emission wavelength shift) sensing mechanisms. I recommend that this section be discussed in greater detail to enhance readers’ understanding. Furthermore, the inclusion of relevant citations: “10.1039/D3CP02714G; 10.1039/C5CS00496A; 10.1016/j.ccr.2025.216470; 10.1002/slct.202404525” will provide stronger support and context for these concepts.

The mechanisms behind turn-on, turn-off, and radiometric probes are described in the following section: PET, PCT, FRET, and AIE. We explain how these types of probes are constructed and provide some review references to support our explanation:

  • Lee, M.H.; Kim, J.S.; Sessler, J.L. Small molecule-based ratiometric fluorescence probes for cations, anions, and biomolecules. Chem. Soc. Rev. 2015, 44, 4185–4191. https://doi.org/10.1039/c4cs00280f.
  • Park, S.H.; Kwon, N.; Lee, J.H.; Yoon, J.; Shin, I. Synthetic ratiometric fluorescent probes for detection of ions. Chem. Soc. Rev. 2020, 49, 143–179. https://doi.org/10.1039/c9cs00243j.
  • Afrin, A.; Jayaraj, A.; Gayathri, M.S.; P., C.A.S. An overview of Schiff base-based fluorescent turn-on probes: a potential candidate for tracking live cell imaging of biologically active metal ions. Sens. Diagn. 2023, 2, 988–1076. https://doi.org/10.1039/d3sd00110e

  1. Table 3, which summarizes [UO₂]²⁺ sensors, is presented without a column outlining the probe stability, potential interferences of the reported studies, and the references also appear to be missing. I recommend that the authors include this information, as it would facilitate a clearer understanding of the methods and materials, highlight their respective strengths and limitations, and improve the overall utility of the table for both reviewers and readers.

Indeed, references to table 3 were missing, which we appreciate you bringing to our attention. Since table 3 is quite extensive, we chose to focus on the sensor type, its fluorescence mechanism, optical properties, LOD, and the matrix of measurement. The interference of these probes is directly discussed in the text.

  1. Tables 4 -7 require refinement to match the style and clarity demonstrated in Table 3, as previously recommended. In addition, the discussions related to Tables 4-7 explicitly integrated into the main text for better coherence and reader understanding.

Tables 4 to 7 have undergone a complete redesign to align with Table 3. All tables now exhibit a harmonious layout, featuring the same columns. References are provided for each cited sensor.

  1. The conclusion section is overly lengthy and should be condensed to present the key findings in a clear and concise manner. A shorter, more focused conclusion will enhance readability and ensure that readers can easily grasp the main take home messages of the review.

Based on your advice, the conclusion was divided into three parts: achievements, application, and challenges:

“Fluorescent probes have emerged as versatile tools for detecting toxic and radiotoxic elements relevant to the nuclear industry, including uranium, cesium, strontium, technetium, zirconium, and beryllium. Over the past decades, a wide range of sensing platforms—such as organic ligands, DNAzymes, MOFs, and quantum dots—have been developed. Recent breakthroughs, particularly with DNAzyme-based probes, COFs and AIE systems, have pushed detection limits to the pico- and nanomolar range, enabling highly sensitive monitoring in both environmental and biological matrices.

These advances hold strong promise for real-world applications. Fluorescent probes can support environmental surveillance around nuclear facilities, assist in nuclear waste management, and track contamination from mining activities. In biological systems, several sensors have demonstrated compatibility with complex media, opening opportunities for studying radionuclide bioavailability, toxicity, and biodistribution. Importantly, fluorescence-based approaches provide a clear advantage for detecting alpha- and beta-emitting radionuclides, where radiometric techniques often require extensive sample preparation or lack sufficient sensitivity.

Despite these advances, several challenges remain before fluorescent probes can be reliably deployed in real-world settings. Improving selectivity in complex mixtures, enhancing sensor stability and reusability, and ensuring fluorophore resistance to both photodegradation and radiolysis are critical for long-term performance. Recent studies have highlighted strategies such as protective matrix embedding, the attachment of protective and anti-fading groups, and the optimization of the measurement medium as promising directions [201,202]. Future work should also emphasize the integration of these probes into portable devices, including microfluidic chips or point-of-care diagnostic platforms, while promoting cross-disciplinary research bridging chemistry, materials science, and radiochemistry. Addressing these issues will be essential to translate laboratory progress into robust, field-ready tools for nuclear safety, environmental monitoring, and biomedical applications.”

  1. The authors should clearly articulate how the present review distinguishes itself from previously published reviews on similar topics. A comparative discussion highlighting the novel aspects, unique focus, or updated perspective of this work would help establish its added value to the field. For example, 10.1039/D5AN00683J; 10.1007/s00604-025-07308-5.

Peng et al. (2025) provided a broad survey of nanomaterial-based platforms for radioactive ion detection in seawater and nuclear wastewater, emphasizing environmental monitoring and sensor integration. Pei et al. (2025) focused on luminescent probes more generally, discussing various radionuclides and ligand design for public safety applications. In contrast, the present review narrows the scope to fluorescent probes relevant to the nuclear fuel cycle (U, Cs, Sr, Zr, Tc, Be), with particular attention on their diversity, highlighting different sensor types, properties, matrices, and applications. Addition in the review:

“This review focuses on the diversity of fluorescent probes, highlighting different sensor types, properties, matrices, and applications, and thus complements recent reviews on sensors for radioactive elements.”

  • Peng, R.; Wang, G.; Wang, C.; Zhang, T.; Qin, N.; Li, X.; Yan, S.; Liu, X. Detection of radioactive ions: current status of nanomaterial-based sensors. Microchim. Acta 2025, 192. https://doi.org/10.1007/s00604-025-07308-5.
  • Pei, Y.; Fang, L.; Zhang, J.; Wang, Z.; Zheng, L.; Zhang, Y.; Zhu, J.; Wang, Z.; Zhang, C.; Pan, J.B. Advancing environmental safety and public health: a comprehensive review of luminescent probes for radioactive element detection. The Analyst 2025. https://doi.org/10.1039/d5an00683j.